# Dietary Supplementation with Eugenol Nanoemulsion Alleviates the Negative Effects of Experimental Coccidiosis on Broiler Chicken’s Health and Growth Performance

**DOI:** 10.3390/molecules28052200

**Published:** 2023-02-27

**Authors:** Mohammad Reza Youssefi, Rahmatollah Alipour, Zahra Fakouri, Mohammad Hassan Shahavi, Nadia Taiefi Nasrabadi, Mohaddeseh Abouhosseini Tabari, Giuseppe Crescenzo, Claudia Zizzadoro, Gerardo Centoducati

**Affiliations:** 1Department of Veterinary Parasitology, Babol Branch, Islamic Azad University, Babol 4747137381, Iran; 2Young Researchers and Elite Club, Babol Branch, Islamic Azad University, Babol 4747137381, Iran; 3Faculty of Engineering Modern Technologies, Amol University of Special Modern Technologies (AUSMT), Amol 4615664616, Iran; 4Department of Parasitology, Karaj Branch, Islamic Azad University, Karaj 3149968111, Iran; 5Faculty of Veterinary Medicine, Amol University of Special Modern Technologies (AUSMT), Amol 4615664616, Iran; 6Department of Veterinary Medicine, University of Bari, 70010 Valenzano, Italy

**Keywords:** phytogenic, coccidiosis, broilers, oxidative stress, inflammation, feed additives

## Abstract

The present study investigated the protective efficacy of dietary supplementation with clove essential oil (CEO), its main constituent eugenol (EUG), and their nanoformulated emulsions (Nano-CEO and Nano-EUG) against experimental coccidiosis in broiler chickens. To this aim, various parameters (oocyst number per gram of excreta (OPG), daily weight gain (DWG), daily feed intake (DFI), feed conversion ratio (FCR), serum concentrations of total proteins (TP), albumin (ALB), globulins (GLB), triglycerides (TG), cholesterol (CHO) and glucose (GLU), serum activity of superoxide dismutase (SOD), glutathione s-transferase (GST), and glutathione peroxidase (GPx)] were compared among groups receiving CEO supplemented feed (CEO), Nano-CEO supplemented feed (Nano-CEO), EUG supplemented feed (EUG), Nano-EUG supplemented feed (Nano-EUG), diclazuril supplemented feed (standard treatment, ST), or basal diet [diseased control (d-CON) and healthy control (h-CON)), from days 1–42. Chickens of all groups, except h-CON, were challenged with mixed *Eimeria* spp. at 14 days of age. Coccidiosis development in d-CON was associated with impaired productivity (lower DWG and higher DFI and FCR relative to h-CON; *p* < 0.05) and altered serum biochemistry (decreased TP, ALB, and GLB concentrations and SOD, GST, and GPx activities relative to h-CON; *p* < 0.05). ST effectively controlled coccidiosis infection by significantly decreasing OPG values compared with d-CON (*p* < 0.05) and maintaining zootechnical and serum biochemical parameters at levels close to (DWG, FCR; *p* < 0.05) or not different from (DFI, TP, ALB, GLB, SOD, GST, and GPx) those of h-CON. Among the phytogenic supplemented (PS) groups, all showed decreased OPG values compared with d-CON (*p* < 0.05), with the lowest value being measured in Nano-EUG. All PS groups showed better values of DFI and FCR than d-CON (*p* < 0.05), but only in Nano-EUG were these parameters, along with DWG, not different from those of ST. Furthermore, Nano-EUG was the only PS group having all serum biochemical values not different (or even slightly improved) relative to ST and h-CON. In conclusion, the tested PS diets, especially Nano-EUG, can limit the deleterious effects of coccidiosis in broiler chickens, due to anticoccidial activity and possibly their reported antioxidant and anti-inflammatory properties, thereby representing a potential green alternative to synthetic anticoccidials.

## 1. Introduction

Health in poultry is threatened by a number of pathogens, among which the protozoan genus of *Eimeria* is one of the most important. Symptoms of infection with *Eimeria* includes malabsorption, enteritis, and, for some *Eimeria* species in severe cases, mortality, compromising birds’ welfare and economic productivity [1]. The costs of treatment for this infection, along with the losses caused by it, are estimated to reach more than EUR 3 billion per year throughout the world [2]. Due to the ever-increasing need of human beings for protein resources from poultry, this pathogenic protozoan and its deleterious effects on productivity will be a big challenge to food security and the global agro-economy [2]. 

Available control measures to limit the hazard of *Eimeria* mainly relies on prophylaxis with anticoccidial drugs and live vaccines. Nevertheless, obstacles exist in each of these two strategies. The major impediment to achieving coccidiosis management through poultry vaccination is the high cost of attenuated vaccine production, which is not economical particularly in developing countries [3]. Meanwhile, mass use of anticoccidials has resulted in drug resistant *Eimeria* species, which—alongside escalating demands on residue-free poultry products—are the main limitations on applying anticoccidials as an efficacious control measure [4,5]. 

Phytogenic feed additives (PFAs) are originated from botanical sources and have gained much interest as cost-effective additives with positive effects on broiler chickens’ immunity, functionality, and health [6]. Several studies have examined the beneficial effects of PFAs as alternatives to synthetic anticoccidials [7,8,9,10,11], but no information is currently available regarding the potential anticoccidial efficacy of clove.

Clove (*Syzygium aromaticum*) is one of the PFAs that has received much attention in poultry farming [12,13], particularly in the form of essential oil [12,14]. The major component of clove essential oil (CEO) is the phenolic compound eugenol (EUG; 4-allyl-2-metthoxyphenol) (75–95%), which is responsible for most of CEO’s bioactivities [15]. The second major active component of CEO could be either β-caryophyllene or eugenol acetate, depending on the specific plant parts (buds, leaves, or stems) from which the oil is extracted [15]. Both CEO and EUG are known to possess, among others, antioxidant, anti-inflammatory, and antibacterial properties [12,16,17,18]; moreover, they have been shown to act as appetite and digestion stimulants, as well as to positively influence intestinal microflora and intestinal mucosal barrier integrity and immune functions in poultry [12,16,18,19]. These biological activities likely account for the efficacy that dietary supplementation with either CEO or EUG has shown in enhancing broiler growth performance under normal conditions (i.e., in the absence of any stressors) [12,19,20], as well as in protecting broilers’ productivity and health against the negative effects of stressors such as heat stress [16] and intestinal pathogenic bacteria such as *Salmonella typhimurium* [17]. In addition, it has been recently reported that in-feed supplementation with EUG formulated as nanoemulsion was able to improve growth performance parameters of broiler chickens even under the challenge of an infection with an avian pathogenic *Escherichia coli* strain [18].

With respect to the latter point, the merit of nanotechnology deserves special mention. Nanotechnology, involving synthesis and development of materials at nanometric scale and their application as diagnostic and therapeutic agents, is a relatively new science field in veterinary medicine and animal production [21,22,23]. Studies have reported performance-enhancing as well as antimicrobial properties for several nano materials in chickens [24,25,26]. Scientific evidence exists showing that nano engineering could improve stability, delivery, and cellular uptake of nutrients and bioactive compounds, by protecting them from the stomach environment, releasing them in an intestinal environment, and consequently increasing their absorption and bioavailability [22]. Nouri (2019) has reported that chitosan nanoencapsulation upgrades beneficial effects of mint, cinnamon, and especially thyme essential oils in broilers. Indeed, dietary supplementation of nanoencapsulated essential oils raised body weight and decreased feed conversion ratio in treated Ross 308 chicks [27].

Bearing the above in mind, the present study aimed to investigate the protective efficacy of CEO and its main constituent EUG, used as PFAs in both free form and nano-formulated emulsions (Nano-CEO and Nano-EUG), against the experimental coccidiosis induced by mixed strains of *Eimeria* spp in broiler chickens. To this purpose, oocyst counting in the excreta was performed as a measure of intensity of infection, some zootechnical data were recorded as measures of productive performance, and some serum biochemical parameters (including antioxidant enzyme activity) were measured as markers of health status. Dietary supplementation with the synthetic anticoccidial drug diclazuril (standard treatment) was used as the reference comparator.

## 2. Results

### 2.1. CEO Chemical Constituents

Results of gas chromatography-mass spectrometry (GC-MS) analysis confirmed eugenol (82.2%) followed by eugenol acetate (12.2%) as the major constituents of the CEO (Appendix A).

### 2.2. Particle Size, Polydispersity INDEX, and Zeta Potential

Mean droplet size in the prepared Nano-CEO and Nano-EUG was 91.91 and 97.31 nm, respectively. The polydispersity index (PDI) was equal to 0.190 and 0.183 for Nano-CEO and Nano-EUG, respectively (Figure 1A,B). The zeta potential of the developed Nano-CEO and Nano-EUG demonstrated surface charge values of −26.2 and −23.9 mV, respectively (Figure 1C,D). These results indicated that the formulated nanoemulsions were on the nanometric scale, and the particles were dispersed homogenously.

### 2.3. Coccidian Oocyst Counting

A significant effect of “treatment” [F _(6, 28)_ = 46.12, *p* = 0.001] and “time” [F _(3, 24)_ = 14.62, *p* = 0.001] was noted on the number of oocysts per gram of excreta (OPG). As could be expected, no oocysts were found in the excreta samples from healthy control (h-CON) group, whereas inoculation of the mixed-strain oocysts resulted in the successful induction of infection in all of the challenged groups, as indicated by detection of oocysts in their excreta on day 21 (D21; i.e., one-week post-challenge) (Figure 2). Monitoring the number of OPG in each of the challenged groups during the subsequent weeks of the study showed that the highest OPG value was achieved on D28 (*p* < 0.05 vs. D21, D35 and D42), followed by a reduction on D35 (*p* < 0.05 vs. D21), and a further reduction on D42 (*p* < 0.05 vs. D21, D28 and D35) (Figure 2). In comparison with the diseased control (d-CON) group, standard treatment (ST) group was characterized by significantly lower OPG at all sampling times (*p* < 0.05). Significantly reduced OPG values relative to d-CON also were observed in all of the phytogenic supplemented groups, although in no case OPG was as low as in the ST group. Moreover, differences were present among the phytogenic treatments with respect to their efficacy at reducing OPG. Particularly, dietary supplementation with Nano-EUG was the most efficacious and resulted in the lowest OPG values in comparison with the other PFAs tested (*p* < 0.05). Nano-CEO and EUG showed intermediate efficacy, with no significant difference being noted between them at any sampling times (*p* > 0.05). Finally, CEO was the least effective phytogenic in controlling oocyst shedding, as indicated by the higher number of OPGs counted in the CEO group relative to the other phytogenic-supplemented groups (*p* < 0.05).

### 2.4. Zootechnical Records

Results of the productive performance evaluation are shown in Table 1. As for average daily weight gain (ADWG), no significant differences were observed in the values of this parameter among experimental groups during the first two weeks of the study (i.e., in the absence of any coccidiosis challenge), with the only exception of a slight decrease recorded in the CEO group on the first week and in the CEO and EUG groups on the second week (*p* < 0.05). From the third week to the end of the study, challenge with coccidiosis in the d-CON group was associated with significantly lower ADWG in comparison with h-CON (*p* < 0.05). Dietary supplementation with ST allowed the maintenance of ADWG values close to those recorded in h-CON, and dietary supplementation with Nano-EUG influenced ADWG in a way that was not different from ST (*p* > 0.05). ADWG values recorded in the other phytogenic supplemented groups showed no significant difference in comparison with d-CON by the end of the study (*p* > 0.05). As for average daily feed intake (ADFI), its values did not differ significantly among the various experimental groups during the first two weeks of the study (*p* > 0.05). Challenge with coccidiosis in d-CON caused significant increase in ADFI values compared with h-CON (*p* < 0.05). In the ST group, this increase was significantly less marked during the time periods of D15-D21 and D22-D28 (lower ADFI compared to d-CON; *p* < 0.05); then, during the last two weeks of the study, ADFI values in this group were not different from those recorded in the h-CON group. Of the four phytogenic supplemented diets, all proved able to limit the coccidiosis-induced increase in ADFI by the end of the study (lower values compared with d-CON; *p* < 0.05), but the values recorded for this parameter in the Nano-EUG group were consistently lower than those of all the other phytogenic supplemented groups (*p* < 0.05), and during the last two weeks of the study, not different from those recorded in ST (and h-CON) group (*p* > 0.05). In line with the abovementioned variations that occurred in ADWG and ADFI values from the third week to the end of the study, feed conversion ratio (FCR) was significantly increased in the coccidiosis-challenged chickens of the d-CON group as compared with h-CON (*p* < 0.05). The ST group showed significantly less marked increase in this parameter (lower value; *p* < 0.05) in comparison with d-CON. A significant improvement (i.e., decrease) of feed conversion ratio (FCR) also was noted in all of the four phytogenic supplemented groups (CEO, Nano-CEO, EUG, and Nano-EUG) relative to d-CON (*p* < 0.05), but only in the Nano-EUG group was the extent of the improvement similar to that recorded in the ST group (*p* > 0.05).

### 2.5. Serum Biochemistry

Biochemical analysis of sera samples at the end of the study (day 42) showed a significant effect of “treatment” on the concentrations of total proteins (TP) [F _(6, 49)_ = 17.29, *p* = 0.001], albumin (ALB) [F _(6, 49)_ = 11.74, *p* = 0.001], and globulins (GLB) [F _(6, 49)_ = 13.15, *p* = 0.001]. More specifically, in the coccidiosis-challenged chickens of the d-CON group, TP, ALB, and GLB concentrations were significantly decreased in comparison with the h-CON group (*p* < 0.05) (Figure 3A–C). In the ST group, this decrease did not occur, and values of all these parameters were not different from those of h-CON (*p* > 0.05). Similarly, in the chickens that received Nano-EUG supplemented diets, the concentrations of ALB and GLB did not differ from h-CON (*p* > 0.05), and those of TP were even slightly higher. As for the other PFAs, EUG proved as effective as ST in preventing the coccidiosis-induced decrease in TP and GLB concentrations (values not significantly different from h-CON; *p* > 0.05), but it had no effect on the coccidiosis-induced decrease in ALB (value not significantly different from d-CON; *p* > 0.05). Nano-CEO was as effective as ST at preventing the coccidiosis-induced decrease in GLB concentrations (values not significantly different from h-CON; *p* > 0.05), but it had only a small and statistically nonsignificant limiting effect on the coccidiosis-induced decrease in TP and ALB (values not different from d-CON; *p* > 0.05). Finally, CEO showed no efficacy at preventing the coccidiosis-induced decrease in any of these protein-related serum biochemical parameters (values not different from d-CON; *p* > 0.05). “Treatment” had no significant effect on serum levels of triglycerides (TG) [F _(6, 49)_ = 1.28, *p* = 0.28], cholesterol (CHO) [F _(6, 49)_ = 2.65, *p* = 0.066], and glucose (GLU) [F _(6, 49)_ = 1.03, *p* = 0.41] (Figure 4A–C, whereas it significantly influenced serum activities of superoxide dismutase (SOD) [F _(6, 49)_ = 3.60, *p* = 0.005], glutathione s-transferase (GST) [F _(6, 49)_ = 25.04, *p* = 0.001], and glutathione peroxidase (GPx) [F _(6, 49)_ = 26.21, *p* = 0.001]. Particularly, in the coccidiosis-challenged chickens of the d-CON group, the activities of all of the three antioxidant enzymes were significantly decreased in comparison with the h-CON group (*p* < 0.05) (Figure 5A–C). In contrast, the ST group showed enzymes activities not different from h-CON (*p* > 0.05). Among the four groups receiving phytogenic supplemented diets, Nano-EUG, similar to ST, was not different from h-CON for any of the enzyme activities measured. Nano-CEO also showed GST and GPx activities not different from h-CON, but its SOD activity remained statistically not different from that of d-CON. In EUG, only GPx activity was not different from h-CON, while GST and SOD activities were significantly less decreased than in d-CON (*p* < 0.05) and statistically not different from d-CON, respectively. Finally, in CEO, no differences were observed in comparison with d-CON, except for a significantly less marked decrease in GPx activity (*p* < 0.05) (Figure 5C).

## 3. Discussion

Recently, PFAs have gained much attention as novel strategies for improving chickens’ health and efficiency in challenges with coccidiosis. They can serve as cost-effective alternatives for more efficient management of avian coccidiosis, while limiting food safety concerns [6]. The most considerable advantage of PFAs relative to synthetic anticoccidials relies in their multiple biological properties, which include interaction with the gut system, in vivo antioxidants, and immunomodulatory activities, along with antimicrobial activity [6,28].

The number of OPGs in excreta is a key parameter for determining anticoccidial efficacy in chickens [29,30]. In the current study, dietary supplementation with each of the four phytogenics tested reduced the number of OPGs in comparison with the coccidiosis-challenged chickens who received the basal diet without supplementation (d-CON), suggesting direct anticoccidial activity. However, none of the phytogenics proved as effective as diclazuril at reducing OPG in chickens. Among the phytogenic supplemented groups, Nano-EUG showed the greatest anticoccidial activity, followed by Nano-CEO, EUG, and CEO. This finding seems to confirm the expectation that nanoformulation could increase the anticoccidial activity of CEO and EUG. Consistently, it has been demonstrated that maduramicin loaded nanostructure lipid carriers have enhanced anticoccidial activity against *E. tenella* in broiler chickens [31], and that toltrazuril-loaded nanocapsules reduced the lesion scores and oocyst excretion in broilers with experimental coccidiosis at half of the dose of the reference toltrazuril [32]. 

A great part of coccidiosis’s economic losses is associated with performance impairments, including decreased weight gain and poor feed conversion ratio [33]. There are several reports on the protective effects of PFAs on the zootechnical performance of coccidiosis-challenged broilers [34,35]. For instance, in the study by Ali et al. (2019), it has been demonstrated that dietary supplementation of garlic and ginger at the rate of 2.5 and 7.5 g/kg in feed significantly improved feed intake, body weight, and FCR in Hubbard broiler chicks with coccidiosis [35]. On the contrary, Galli et al. (2021) have reported no beneficial effect of adding PFA blend composed of curcuminoids, cinnamaldehyde, and glycerol monolaurate on productive performance of broilers challenged with coccidiosis [36]. Probably, differences in the bioactive compounds’ composition, dose of supplementation, and species of *Eimeria*, which have been used for coccidiosis induction, are among the contributing factors causing variations in the clinical efficacy of different PFAs [37]. All of the PFAs tested in the present study proved able to limit the coccidiosis-dependent impairment of at least two of the zootechnical parameters measured (ADFI and FCR), but only Nano-EUG showed efficacy comparable to that of the standard treatment with diclazuril in maintaining the values of the zootechnical parameters at levels close to (ADWG, FCR) or not different from (ADFI) those measured in the healthy (nonchallenged) chickens (h-CON). It is worth noting that this similarity in the extent of the protective efficacy exerted by ST and Nano-EUG on the productive performance of coccidiosis-challenged chickens occurred in spite of the lower anticoccidial activity exhibited by the former relative to the latter treatment (i.e., smaller reduction of OPG; see above).

Similarly, in the present study, dietary supplementation with Nano-EUG was found to be as effective as that with diclazuril at maintaining serum concentrations of TP, ALB, and GLB, as well as serum activities of the antioxidant enzymes SOD, GST, and GPx in the coccidiosis-challenged chickens within levels not different from those measured in chickens of the h-CON group. On the whole, these findings suggest that, besides direct anticoccidial activity, other biological properties are probably involved in the ability of Nano-EUG to mitigate the severity of coccidiosis and thereby preserve the health status and productive performance of the infected chickens. In this regard, several studies have reported antioxidant and anti-inflammatory effects for CEO, EUG, and their nanoformulations [14,16,17,18,38,39,40,41]. Coccidiosis in chickens involves oxidative stress and reduces antioxidant enzymes including SOD, catalase, and GPx [42], which is also demonstrated in the present study. Moreover, oxidative stress and inflammation are interconnected phenomena, as reactive oxygen species that are not adequately scavenged by enzymatic and nonenzymatic antioxidant systems cause cellular damage [43]. So, it seems plausible that a contribution to the ability of the Nano-EUG-supplemented diet to limit the deleterious consequences of coccidiosis could come from modulation of the antioxidant status and reduction of inflammatory mediators mainly in the gut system of the challenged chickens.

## 4. Materials and Methods

### 4.1. Phytochemicals

CEO was purchased from a local pharmaceutical company (Tehran, Iran). The chemical composition of CEO was verified by analysis on an Agilent 7890A gas chromatograph (GC) equipped with 5975C mass spectrometer (MS) equipped with a HP column of 5 m long (0.25 mm diameter and 0.25 cm internal diameter). The carrier gas was helium at a flow rate of 1 mL/min. The analytical conditions were initial temperature of 100 °C (increasing 8 °C per minute until final temperature of 250 °C); inlet temperature and mass detector were 250 °C and 300 °C, respectively. The oil sample was diluted to 1% with n-hexane, and 2 µL of the solution was injected into the GC-MS system. The identification of the compounds was based on the comparison of the retention indices and mass spectra with those contained in the commercial libraries. EUG (E51791) was purchased from Sigma-Aldrich (Germany).

### 4.2. Nanoemulsion Development and Characterization

Nano-CEO and Nano-EUG were developed by applying the ultrasonication emulsification method. More specifically, the formulation included 15% (*w*/*w*) CEO or EUG, as well as 5% (*w*/*w*) of the mixture of Span 80 and Tween 80 surfactants at a weight ratio of 56:44% with hydrophilic lipophilic balance (HLB) equal to 9; the remainder consisted of MilliQ water. All ingredients were mixed at room temperature (25 °C) and agitated for 3 min on the vortex mixer to obtain the initial emulsion. Finally, the samples were placed in an ultrasonic UP400S processor (Hielscher Ultrasonics GmbH, Teltow, Germany) and irradiated for 10 min (24 kHz and 0.8 w/cm^2^). Ultrasonic waves mixed the phases more uniformly and in smaller droplet sizes [44]. The particle size, zeta potential, and PDI of the prepared nanoemulsions were determined by the Nano-ZS ZEN analyzer (Malvern, UK).

### 4.3. Animals and Study Design

One hundred and five one-day-old Ross 308 broiler chickens with average weight of 45.2 ± 0.5 g were kept in wire cages according to the battery efficacy test protocol [45]. Birds had ad libitum feed and water access, and the diets were formulated according to Ross 308 recommendations. To produce the phytogenic supplemented diets, the tested phytochemicals (CEO and EUG) and their nanoformulations were added to the ingredients of the basal diet at 100 mg/kg complete feed and formulated in mash form. Details of the basal diet composition are listed in Appendix A.

The birds were randomly allocated into the 7 following groups (each having 3 replicates of 5 birds): (a) CEO, which consisted of chickens receiving CEO at 100 mg/kg of feed; (b) Nano-CEO, which consisted of chickens receiving Nano-CEO at 100 mg/kg of feed; (c) EUG, which consisted of chickens receiving EUG at 100 mg/kg of feed; (d) Nano-EUG, which consisted of chickens receiving Nano-EUG at 100 mg/kg of feed; (e) ST, which consisted of chickens receiving standard treatment with diclazuril (Aras Bazar Pharmaceutical Company, Amol, Iran) at 1 mg/kg of feed (the manufacturer’s suggested dose; and (f) diseased-control (d-CON) and g) healthy control (h-CON), which both consisted of chickens receiving only basal diet without any supplementation. The selected level of diet supplementation was consistent with inclusion levels of CEO and EUG that previous studies found to be effective in improving broiler chickens’ growth performance [19,20]. Birds received either basal or supplemented diets from days 1–42 (namely, for 6 weeks). All of the birds, except those of the h-CON group, were challenged with coccidiosis at 14 days of age (i.e., at the end of week 2). In order to mimic the condition that most often occurs in the field, experimental coccidiosis was induced by inoculation of 1 mL mixed sporulated oocysts of four pathogenic species of *Eimeria* including 8 × 10^4^ *E. tenella*, 1 × 10^4^ *E. necatrix*, 5 × 10^3^ *E. acervuline*, and 5 × 10^3^ *E. maxima* [30]. The study was authorized by the Institutional Ethics Committee of Animal Care and Use with approval number IR.IAU.BABOL.REC.1401.021, in accordance with animal welfare following regulations under the Animals (Scientific Procedures) Act 1986 (ASPA).

### 4.4. Oocyst Counting: Sample Collection and Analysis

For coccidian oocyst counting, used as a measure of infection intensity, feces were sampled on a weekly basis post experimental infection (i.e., at the end of week 3 (day 21—D21), 4 (day 28—D28), 5 (day 35—D35), and 6 (day 42—D42) of the study), and number of oocysts per gram of excreta (OPG) was determined by the Mc Master technique [29].

### 4.5. Zootechnical Records

The zootechnical data used as measures of productive performance included average daily weight gain (ADWG), average daily feed intake (ADFI), and feed conversion ratio (FCR). ADWG and ADFI were calculated based on the records of body weight and feed consumption that were kept for each experimental group at the end of each week of the study period; FCR was calculated by dividing the weekly records of ADFI and ADWG.

### 4.6. Serum Biochemical Parameters: Sample Collection and Analysis

For measurement of the serum biochemical parameters used as markers of animal health status, blood samples from eight birds of each experimental group were collected at the end of week 6 (i.e., at the end of the study) by brachial wing venipuncture. After centrifugation at 3500 rpm for 10 min, sera were separated and stored at −20 °C for further analysis. The selected biochemical parameters included: concentrations of total proteins (TP), albumin (ALB), triglycerides (TG), cholesterol (CHO), and glucose (GLU), which were measured using commercial kits (Pars Azmun Co. Ltd., Tehran, Iran) according to manufacturer’s instructions and by means of an autoanalyzer (Mindray BS200, China); concentration of globulins (GLB), which was calculated by subtraction of ALB from TP; and activities of the antioxidant enzymes superoxide dismutase (SOD), glutathione s-transferase (GST) and glutathione peroxidase (GPx), which were measured spectrophotometrically by commercial kits (Navand Salamat, Iran) used according to the procedures of their manufacturer.

### 4.7. Statistical Analysis

The obtained data were subjected to Kolmogorov–Smirnov test for checking normality. For zootechnical data, the comparison between different experimental groups was conducted using the Wilcoxon signed-rank test with a Bonferroni adjustment, whereas the OPG data were analyzed with Friedman repeated measure analysis of variance (factors: “treatment” and “time”). Serum biochemistry data was subjected to one-way analysis of variance (ANOVA) and Bonferroni’s as the post-hoc test. Values of *p* < 0.05 were considered statistically significant. The data were analyzed by SPSS version 26 (Chicago, USA).

## 5. Conclusions

Overall, the PFAs tested in the present study, especially nanoformulated EUG, showed notable protective effects on chickens’ performance and health status against coccidiosis, and may therefore represent a promising alternative to synthetic coccidiostats for management of this parasitic disease in broiler chickens. Considering the “generally recognized as safe” (GRAS) status of EUG [46], the findings reported herein may provide a valuable contribution to the development of green approaches in sustainable farming to gain organic and residue-free poultry products.

## Figures and Tables

**Figure 1 molecules-28-02200-f001:**
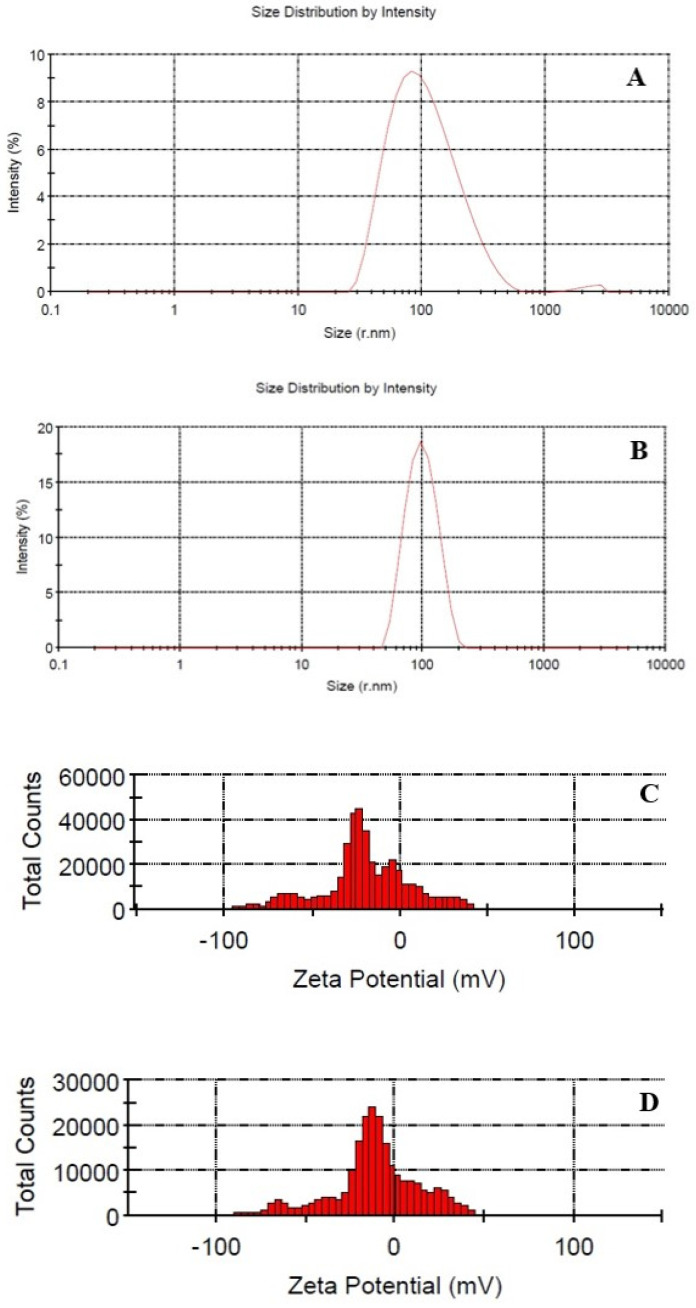
Particle size and polydispersity index distribution of nanoemulsion of clove essential oil (Nano-CEO) (**A**) and nanoemulsion of eugenol (Nano-EUG) (**B**); zeta potential for Nano-CEO (**C**) and Nano-EUG (**D**).

**Figure 2 molecules-28-02200-f002:**
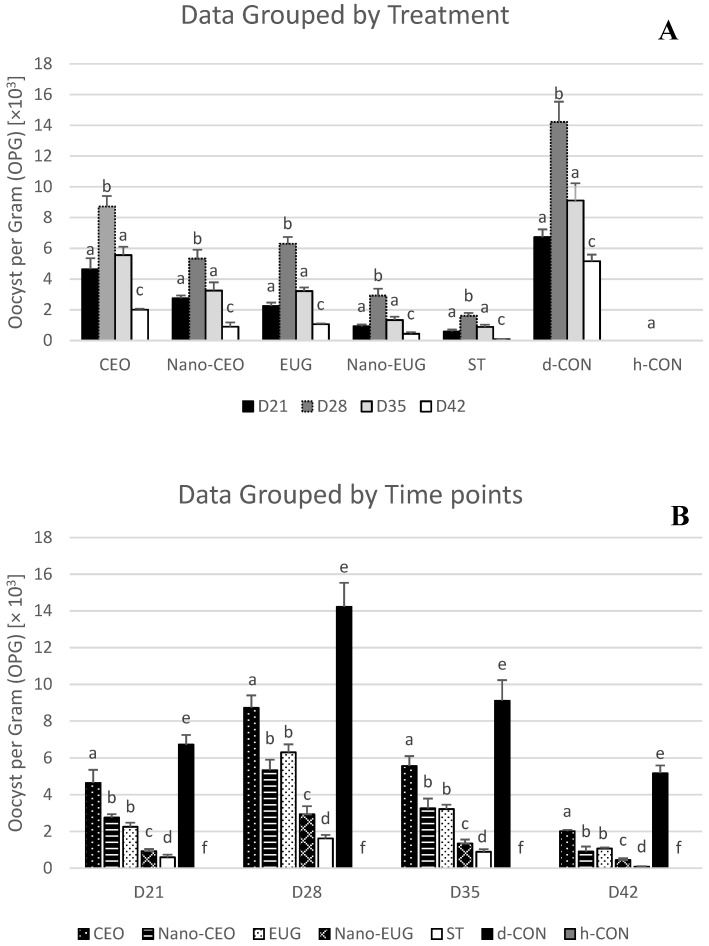
Mean number of oocyst per gram of excreta (OPG) [×10^3^] grouped by treatments (**A**) and time points (**B**) counted at the end of weeks 3 (D21), 4 (D28), 5 (D35), and 6 (D42) of the study in broiler chickens under the following experimental conditions: CEO (chickens receiving clove essential oil at 100 mg/kg in feed from day 1–42 and challenged with coccidiosis at 14 days of age); Nano-CEO (chickens receiving nanoemulsion of clove essential oil at 100 mg/kg in feed from day 1–42 and challenged with coccidiosis at 14 days of age); EUG (chickens receiving eugenol at 100 mg/kg in feed from day 1–42 and challenged with coccidiosis at 14 days of age); Nano-EUG (chickens receiving nanoemulsion of eugenol at 100 mg/kg in feed from day 1–42 and challenged with coccidiosis at 14 days of age); ST (chickens receiving standard treatment with diclazuril at 1 mg/kg in feed from day 1–42 and challenged with coccidiosis at 14 days of age); d-CON (chickens serving as diseased controls, i.e., receiving basal diet without any supplementation from day 1–42 and challenged with coccidiosis at 14 days of age); and h-CON (chickens serving as healthy controls, i.e., receiving basal diet without any supplementation from day 1–42 and not challenged with coccidiosis). Data are presented as means ± SEM. (**A**) Columns marked with different letters are significantly different between time points. (**B**) Columns marked with different letters at each time point are significantly different between experimental conditions.

**Figure 3 molecules-28-02200-f003:**
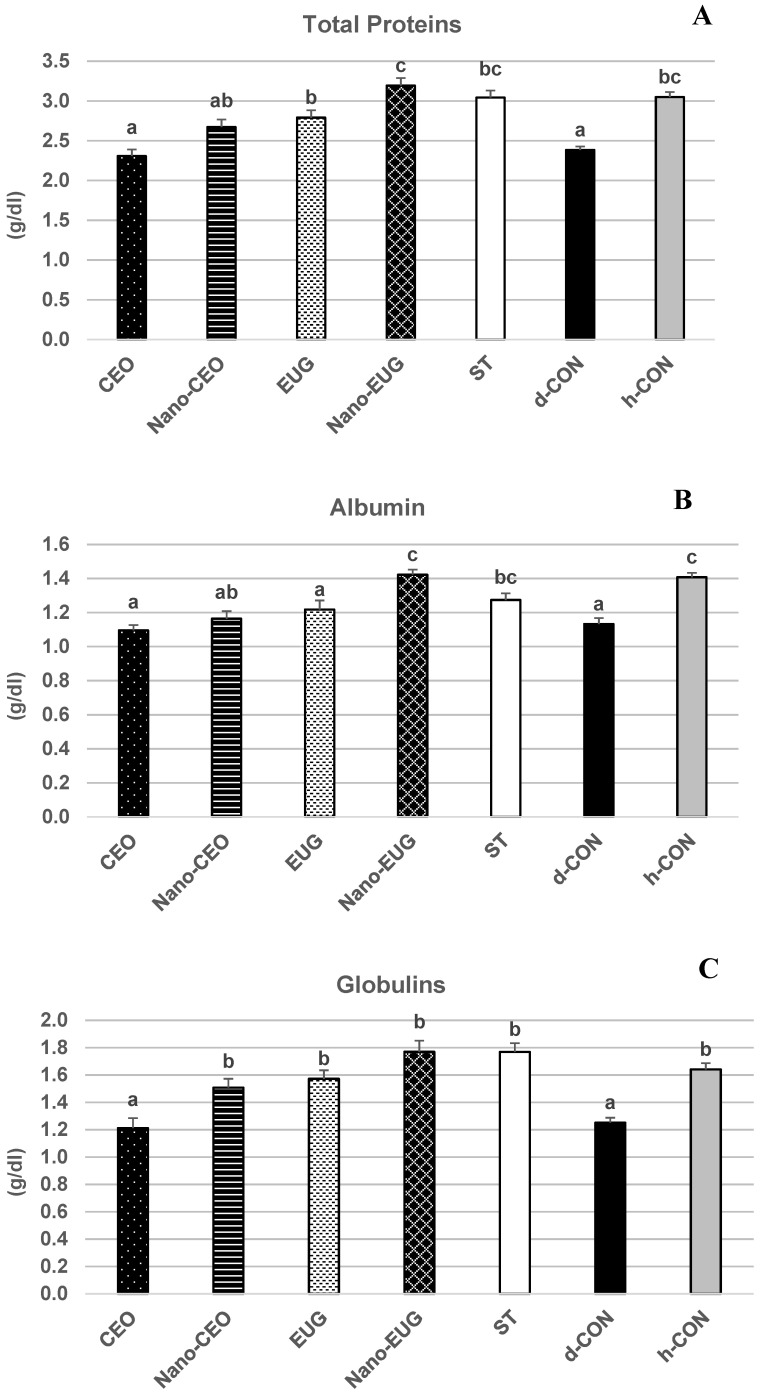
Serum concentrations of total proteins (**A**), albumin (**B**) and globulins, (**C**) measured on day 42 in broiler chickens under the following experimental conditions: CEO (chickens receiving clove essential oil at 100 mg/kg in feed from day 1–42 and challenged with coccidiosis at 14 days of age); Nano-CEO (chickens receiving nanoemulsion of clove essential oil at 100 mg/kg in feed from day 1–42 and challenged with coccidiosis at 14 days of age); EUG (chickens receiving eugenol at 100 mg/kg in feed from day 1–42 and challenged with coccidiosis at 14 days of age); Nano-EUG (chickens receiving nanoemulsion of eugenol at 100 mg/kg in feed from day 1–42 and challenged with coccidiosis at 14 days of age); ST (chickens receiving standard treatment with diclazuril at 1 mg/kg in feed from day 1–42 and challenged with coccidiosis at 14 days of age); d-CON (chickens serving as diseased controls, i.e., receiving basal diet without any supplementation from day 1–42 and challenged with coccidiosis at 14 days of age); and h-CON (chickens serving as healthy controls, i.e., receiving basal diet without any supplementation from day 1–42 and not challenged with coccidiosis). Data are presented as means ± SEM; bars that do not share a common letter (a–c) differ significantly.

**Figure 4 molecules-28-02200-f004:**
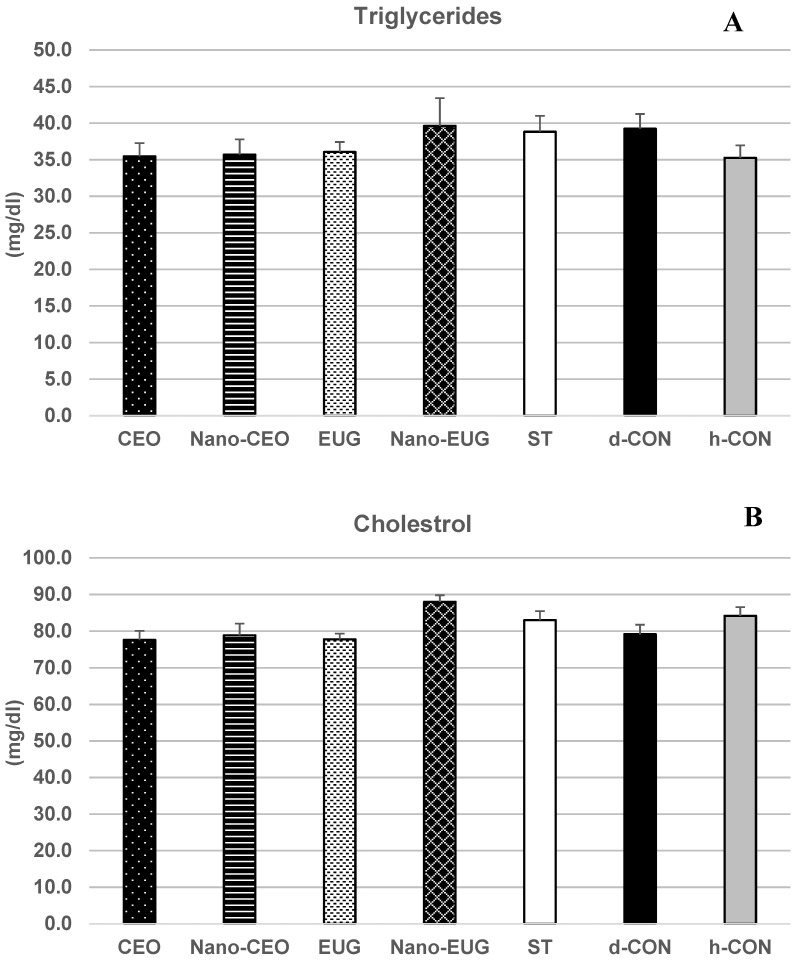
Serum concentrations of triglycerides (**A**), cholesterol (**B**), and glucose (**C**) measured on day 42 in broiler chickens under the following experimental conditions: CEO (chickens receiving clove essential oil at 100 mg/kg in feed from day 1–42 and challenged with coccidiosis at 14 days of age); Nano-CEO (chickens receiving nanoemulsion of clove essential oil at 100 mg/kg in feed from day 1–42 and challenged with coccidiosis at 14 days of age); EUG (chickens receiving eugenol at 100 mg/kg in feed from day 1–42 and challenged with coccidiosis at 14 days of age); Nano-EUG (chickens receiving nanoemulsion of eugenol at 100 mg/kg in feed from day 1–42 and challenged with coccidiosis at 14 days of age); ST (chickens receiving standard treatment with diclazuril at 1 mg/kg in feed from day 1–42 and challenged with coccidiosis at 14 days of age); d-CON (chickens serving as diseased controls, i.e., receiving basal diet without any supplementation from day 1–42 and challenged with coccidiosis at 14 days of age); and h-CON (chickens serving as healthy controls, i.e., receiving basal diet without any supplementation from day 1–42 and not challenged with coccidiosis). Data are presented as means ± SEM; no significant difference was noted between groups.

**Figure 5 molecules-28-02200-f005:**
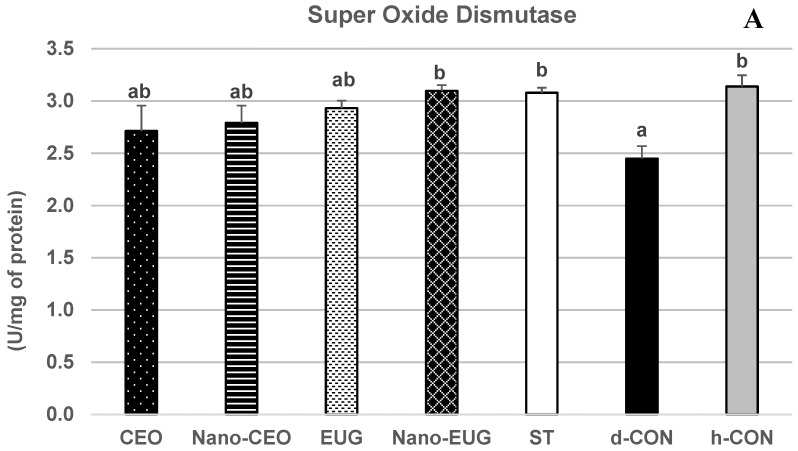
Serum activity of the antioxidant enzymes Super Oxide Dismutase (**A**), Glutathione S-Transferase (**B**), and Glutathione Peroxidase (**C**) measured on day 42 in broiler chickens under the following experimental conditions: CEO (chickens receiving clove essential oil at 100 mg/kg in feed from day 1–42 and challenged with coccidiosis at 14 days of age); Nano-CEO (chickens receiving nanoemulsion of clove essential oil at 100 mg/kg in feed from day 1-42 and challenged with coccidiosis at 14 days of age); EUG (chickens receiving eugenol at 100 mg/kg in feed from day 1–42 and challenged with coccidiosis at 14 days of age); Nano-EUG (chickens receiving nanoemulsion of eugenol at 100 mg/kg in feed from day 1–42 and challenged with coccidiosis at 14 days of age); ST (chickens receiving standard treatment with diclazuril at 1 mg/kg in feed from day 1–42 and challenged with coccidiosis at 14 days of age); d-CON (chickens serving as diseased controls, i.e., receiving basal diet without any supplementation from day 1–42 and challenged with coccidiosis at 14 days of age); and h-CON (chickens serving as healthy controls, i.e., receiving basal diet without any supplementation from day 1–42 and not challenged with coccidiosis). Data are presented as means ± SEM; bars that do not share a common letter (a–d) differ significantly.

**Table 1 molecules-28-02200-t001:** Parameters of productive performance (Average daily weight gain (gram), average daily feed intake (gram), and feed conversion ratio (feed intake/weight gain)) recorded in broiler chickens under the following experimental conditions: CEO (chickens receiving clove essential oil at 100 mg/kg in feed from day 1–42 and challenged with coccidiosis at 14 days of age); Nano-CEO (chickens receiving nanoemulsion of clove essential oil at 100 mg/kg in feed from day 1–42 and challenged with coccidiosis at 14 days of age); EUG (chickens receiving eugenol at 100 mg/kg in feed from day 1–42 and challenged with coccidiosis at 14 days of age); Nano-EUG (chickens receiving nanoemulsion of eugenol at 100 mg/kg in feed from day 1–42 and challenged with coccidiosis at 14 days of age); ST (chickens receiving the standard treatment with diclazuril at 1 mg/kg in feed from day 1–42 and challenged with coccidiosis at 14 days of age); d-CON (chickens serving as diseased controls, i.e., receiving basal diet without any supplementation from day 1–42 and challenged with coccidiosis at 14 days of age); and h-CON (chickens serving as healthy controls, i.e., receiving basal diet without any supplementation from day 1–42 and not challenged with coccidiosis). Data are presented as means ± SEM; in each column, means that do not share a common letter (a–e) differ significantly.

	**Treatments**	**Time Period Pre-Challenge**			**Time Period Post-Challenge**			
		**D1–D7**	**D8–D14**		**D15–D21**	**D22–D28**	**D29–D35**	**D36–D42**
**Average Daily Weight Gain**	CEO	17.20 ± 0.32^a^	29.66 ± 0.89 ^a^		42.56 ± 0.74 ^a^	67.79 ± 1.09 ^ab^	77.58 ± 0.84 ^ab^	95.90 ± 0.80 ^a^
	Nano-CEO	18.26 ± 0.21^b^	32.70 ± 0.40 ^b^		48.86 ± 0.77 ^b^	68.06 ± 1.00 ^ab^	80.72 ± 0.89 ^b^	96.32 ± 0.82 ^a^
	EUG	18.26 ± 0.32 ^ab^	29.18 ± 0.79 ^a^		47.02 ± 0.92^b^	70.26 ± 0.91^b^	81.16 ± 0.71 ^b^	97.28 ± 0.60 ^ab^
	Nano-EUG	18.81 ± 0.22 ^ab^	33.56 ± 0.31 ^b^		54 ± 0.43 ^c^	74.39 ± 0.71 ^bc^	88.30 ± 0.68 ^c^	99.81 ± 0.69 ^b^
	ST	19.02 ± 0.30 ^b^	32.30 ± 0.29 ^b^		52.89 ± 0.38 ^c^	73.28 ± 0.52 ^b^	85.91 ± 0.48 ^c^	100.08 ± 0.43 ^b^
	d-CON	18.24 ± 0.25 ^ab^	32.90 ± 0.47 ^b^		47.97 ± 0.77 ^b^	65.68 ± 0.91 ^a^	76.37 ± 1.04 ^a^	92.90 ± 1.14 ^a^
	h-CON	18.84 ± 0.26 ^b^	33.16 ± 0.39 ^b^		54.96 ± 0.36 ^c^	77.60 ± 0.80 ^c^	89.44 ± 0.76 ^c^	105.36 ± 0.64 ^c^

**Average Daily Feed Intake**	CEO	25.89 ± 0.49 ^a^	49.58 ± 0.44 ^a^		83.38 ± 0.74 ^a^	132.03 ± 1.08 ^a^	167.40 ± 1.38 ^ab^	215.97 ± 1.04 ^a^
	Nano-CEO	25.86 ± 0.36 ^a^	49.97 ± 0.42 ^a^		82.00 ± 0.96 ^ab^	127.42 ± 0.65 ^b^	163.88 ± 1.23 ^b^	213.13 ± 1.03 ^a^
	EUG	25.46 ± 0.38 ^a^	49.72 ± 0.50 ^a^		81.44 ± 0.58 ^ab^	127.86 ± 0.79 ^b^	162.13 ± 1.19 ^b^	212.11 ± 0.74 ^a^
	Nano-EUG	25.66 ± 0.24 ^a^	49.14 ± 0.57 ^a^		75.89 ± 0.55 ^d^	118.68 ± 0.60 ^cd^	156.08 ± 0.69 ^c^	206.67 ± 0.72 ^b^
	ST	25.24 ± 0.31 ^a^	48.95 ± 0.43 ^a^		79.74 ± 0.75 ^bc^	121.22 ± 0.52 ^c^	157.71 ± 1.16 ^c^	207.59 ± 0.97 ^b^
	d-CON	25.67 ± 0.23 ^a^	48.39 ± 0.57 ^a^		87.75 ± 0.93 ^e^	133.26 ± 1.4 ^a^	169.23 ± 1.51 ^a^	221.02 ± 1.18 ^c^
	h-CON	25.97 ± 0.31 ^a^	49.45 ± 0.57 ^a^		77.18 ± 0.70 ^cd^	115.74 ± 1.15 ^d^	153.68 ± 0.98 ^c^	204.93 ± 0.86 ^b^

**Feed Conversion Ratio**	CEO	1.51 ± 0.03 ^a^	1.69 ± 0.06 ^a^		1.96 ± 0.03 ^a^	1.95 ± 0.03 ^a^	2.16 ± 0.02 ^a^	2.25 ± 0.02 ^a^
	Nano-CEO	1.41 ± 0.02 ^ab^	1.53 ± 0.02 ^b^		1.68 ± 0.02 ^b^	1.87 ± 0.02 ^ab^	2.03 ± 0.02 ^bc^	2.21 ± 0.01 ^a^
	EUG	1.40 ± 0.03 ^ab^	1.72 ± 0.05 ^a^		1.74 ± 0.03 ^cb^	1.82 ± 0.03 ^b^	1.99 ± 0.02 ^c^	2.18 ± 0.01 ^a^
	Nano-EUG	1.36 ± 0.02 ^b^	1.46 ± 0.02 ^cb^		1.40 ± 0.01 ^d^	1.59 ± 0.01 ^cd^	1.76 ± 0.01 ^de^	2.07 ± 0.01 ^b^
	ST	1.33 ± 0.02 ^b^	1.51 ± 0.01 ^b^		1.50 ± 0.02 ^d^	1.65 ± 0.01 ^c^	1.83 ± 0.01 ^e^	2.07 ± 0.01 ^b^
	d-CON	1.40 ± 0.01 ^a^	1.47 ± 0.02 ^cb^		1.83 ± 0.04 ^c^	2.03 ± 0.03 ^a^	2.22 ± 0.03 ^ab^	2.38 ± 0.03 ^c^
	h-CON	1.38 ± 0.02 ^b^	1.49 ± 0.02 ^cb^		1.40 ± 0.01 ^d^	1.49 ± 0.01 ^d^	1.72 ± 0.01 ^d^	1.94 ± 0.01 ^d^

## Data Availability

Data of the present study is available upon request from the corresponding authors.

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
