# Peer review of "Dietary Supplementation with Eugenol Nanoemulsion Alleviates the Negative Effects of Experimental Coccidiosis on Broiler Chicken’s Health and Growth Performance"

_molecules, 2023, doi:10.3390/molecules28052200_

Round 1

Reviewer 1 Report

I read with interest the manuscript entitled "Dietary supplementation with Eugenol Nanoemulsion Alleviates the Negative Effects of Experimental Coccidiosis on Broiler 3 Chicken’s health and growth performance". The current study aimed to investigate the protective efficacy of dietary supplementation with clove essential oil (CEO), its main constituent eugenol (EUG), and their nano-formulated emulsions (Nano-CEO and Nano-EUG) against experimental coccidiosis in broiler chickens. In order to achieve the study goal, the authors assessed some biological records, serum biochemical parameters and enzymatic antioxidants. The manuscript is clearly written, and the scientific idea is obviously described. However, there are a few comments.

- Abbreviations (acronyms) not generally known should be followed in the text by the spelled-out forms in parentheses the first time they occur, for example, ADWG.

- Figure 2 is crowded with letters, it should be simplified for better understanding.

- In Table 1, the author should explain whether the used superscript letters represent comparison in the same row or column.

- In the M&M section, for the applied doses of diclazuril, CEO and EUG as well as the analytical methods used to estimate biochemical parameters, references should be added.

- At the end of discussion section, the author concluded that "the ability of the Nano-EUG supplemented diet to limit the deleterious consequences of coccidiosis could come from modulation of the antioxidant status and reduction of inflammatory mediators mainly in the gut system of the challenged chickens". On what basis, did the author build their conclusion while they did not assess any inflammatory mediators or free radicals!

Thank you!

Reviewer 2 Report

A rather interesting study, although not entirely innovative, studies dealing with clove, and its essential oil, in poultry are from 1995.  

This will also be related to my first comment, where I would definitely recommend revise better the introduction between lines 76-81.

The designation of groups is given quite confusingly. ST group is listed for the first time in the results section of line 125 without an explanation of which group it is.

Another very confusing part is the statistical evaluation. In figure 2, it is extremely difficult to navigate what is different from what. It needs to be revised.

In addition, as in all other figures, the statistical significance of the differences is not indicated. Which is a necessity.

If the data in the graphs included in figure 4 do not differ, no letters indicating statistical differences are indicated.

Subchapter 4.1. in Materials and Methods. is insufficient. Define the analysis conditions as well as the composition detected.

Round 2

Reviewer 2 Report

Authors have made the indicated changes in the manuscript, except for two. Figure 2 remains just as confusing as it was when it comes to group comparisons. In figure 4, where they didn't understand what it was about. I don't need to explain what "a" means, but it is completely useless to highlight any letter regarding statistical significance if there is absolutely none.

Author Response

Reviewer 2 Round 2

Comments and Suggestions for Authors

Authors have made the indicated changes in the manuscript, except for two.

Reply: Thank you. We have tried to do the revision according to the comments completely.

Figure 2 remains just as confusing as it was when it comes to group comparisons.

Reply: we revised the figure 2 and displayed the data and comparisons in 2 separate figures (Figure 2 A and B) in order to make it clearer.

As the respected reviewer commented, we showed the significant difference by just one set of lowercase letters in figures, one figure for the between groups comparison and the other for time points comparison. We hope that this revision has made the updated figure 2 acceptable.

In figure 4, where they didn't understand what it was about. I don't need to explain what "a" means, but it is completely useless to highlight any letter regarding statistical significance if there is absolutely none.

Reply: Sorry for this misunderstanding from our side. We deleted the letters in figure 4 and revised the legend.